# Genome Mining Reveals Pathways for Terpene Production in Aerobic Endospore-Forming Bacteria Isolated from Brazilian Soils

**DOI:** 10.3390/microorganisms13112528

**Published:** 2025-11-04

**Authors:** Felipe de Araujo Mesquita, Waldeyr Mendes Cordeiro da Silva, Taina Raiol, Marcelo de Macedo Brigido, Nalvo Franco de Almeida, Bruna Fuga, Danilo de Andrande Cavalcante, Marlene Teixeira De-Souza

**Affiliations:** 1Department of Cellular Biology, Institute of Biological Sciences, University of Brasilia (UnB), Brasilia 70910-900, Brazil; felipedearaujomesquita@gmail.com (F.d.A.M.); brigido@unb.br (M.d.M.B.); bruna.araujo@unb.br (B.F.); danilo.ac9@gmail.com (D.d.A.C.); 2Microbial Biology Graduation Program, University of Brasilia (UnB), Brasilia 70910-900, Brazil; 3Instituto Federal de Goiás, Formosa 73813-816, GO, Brazil; waldeyr.mendes@ifg.edu.br; 4Fiocruz Brasília, Oswaldo Cruz Foundation, Brasilia 70904-130, Brazil; taina.raiol@fiocruz.br; 5Faculdade de Computação, Universidade Federal de Mato Grosso do Sul, Campo Grande 79070-900, MS, Brazil; nalvojr@gmail.com

**Keywords:** *Bacillus* and related genera, secondary or specialised metabolites, the antiSMASH software, biosynthetic gene cluster (BGC), conserved signature indels for different clades, methylerythritol-phosphate (MEP) pathway, lycopene, squalene, hopanoids

## Abstract

Terpenes are the largest category of specialised metabolites. Aerobic endospore-forming bacteria (AEFB), a diverse group of microorganisms, can thrive in various habitats and produce specialised metabolites, including terpenes. This study investigates the potential for terpene biosynthesis in 10 AEFB strain whole-genome sequences by performing a bioinformatics analyses to identify genes associated with these isoprene biosynthesis pathways. Specifically, we focused on the sequences coding for enzymes in the methylerythritol-phosphate (MEP) pathway and the polyprenyl synthase family, which play crucial roles in synthesising terpene precursors together with terpene synthases. A comparative analysis revealed the unique genetic architecture of these biosynthetic gene clusters (BGCs). Our results indicated that some strains possessed the complete genetic machinery required to produce terpenes such as squalene, hopanoids, and carotenoids. We also reconstructed phylogenetic trees based on the amino acid sequences of terpene synthases, which aligned with the phylogenetic relationships inferred from the whole-genome sequences, suggesting that the production of terpenes is an ancestor property in AEFB. Our findings highlight the importance of genome mining as a powerful tool for discovering new biological activities. Furthermore, this research lays the groundwork for future investigations to enhance our understanding of terpene biosynthesis in AEFB and the potential applications of these Brazilian environmental strains.

## 1. Introduction

Specialised or secondary metabolites are not essential for growth. Nevertheless, they play a relevant ecological role by providing nutrients in competitive environments and offering adaptive advantages to the producing organism [1,2]. Terpenes are hydrocarbons of linked five-carbon units of isoprene [3], and they are the largest category of specialised metabolites, with over 80,000 known structures [4]. These molecules present a wide range of applications in pharmaceuticals, food, and cosmetics, among other relevant industries [3,4].

Terpenes are fundamental in the protection against diseases in plants. This property resulting from the antimicrobial activity of these molecules prevents the reproduction and development of phytopathogenic microorganisms [5]. In addition, these specialised metabolites can act as hormones, protective pigments responsible for the colours of various tissues, and odours, which protect plants against herbivory [6,7]. The social and bioeconomic importance of terpenes has increased their relevance in the pharmaceutical and cosmetics industries. Terpenes also play a prominent role in human health due to their excellence as antitumor, antimicrobial, anti-allergic, anti-inflammatory, and antioxidant agents, among other applications [5,8]. Likewise, terpenes are suitable for the food and beverage industries as preservatives or aromatic additives [9]. Additionally, they are recognised as possible energy-transition agents in biofuel formulations.

Terpenes were believed to be produced exclusively by plants, but it is now understood that their synthesis occurs in bacteria, fungi, protozoa, and invertebrates [6,10,11,12]. Various biochemical pathways are involved in terpene production, which can be independent or integrated [13]. The synthesis of these isoprenes starts from the precursor isopentenyl pyrophosphate (IPP), produced via either the mevalonate (MVA) or the methylerythritol-phosphate (MEP) pathways [14].

The enzyme isopentenyl pyrophosphate isomerase (IDI) converts IPP into dimethylallyl pyrophosphate (DMAPP), an IPP isomer. Subsequently, geranyl pyrophosphate synthase (GPPS) condenses these two isomers to produce geranyl pyrophosphate (GPP), which serves as monoterpenes’ precursor. GPPS belongs to the enzyme family polyprenyl synthase (PPS), which includes farnesyl pyrophosphate synthase (FPPS) and geranyl-geranyl pyrophosphate synthase (GGPPS). FPPS is responsible for sesquiterpenes’ and triterpenes’ syntheses, and GGPPS is involved in diterpene and tetraterpene precursors. An overview of the substrates for the MEP pathway, the enzymes of the PPS family, and their corresponding products is provided in Table 1.

Plants and fungi utilise the MVA and MEP pathways for terpene synthesis. Indeed, both pathways are involved in the biosynthesis of gibberellins, a diterpene plant hormone class, crucial for various developmental processes in *Arabidopsis thaliana* [15]. Pardo et al. [16] expressed a geraniol synthase gene from *Ocimum basilicum* (sweet basil) in a *Saccharomyces cerevisiae* wine strain. This construction changed the terpene profile of wine, because these self-aromatising recombinant yeasts overproduce these plant metabolites in wines de novo.

Specialised metabolites exhibit diverse biological activities within fungal communities, which play vital roles in adaptation, defence, competition, and communication [17]. Comparative genomics and metabolomics were used to investigate the high diversity of terpene BGCs potentially involved in the fungal competition and communication of three *Suillus* species [18]. Though the functionality of these BGCs could not be understood adequately during this study, these authors observed that terpenes were significantly more abundant in co-culture conditions. The mycelium of wood-rotting fungi *Polyporus brumalis* produces two sesquiterpenes by the MVA and MEP pathways [19]. Even if the co-expression of genes involved in these pathways can help elucidate these sesquiterpenes’ synthesis, these observations do not fully explain the production mechanisms.

Most bacteria natively possess the MEP pathway for terpene synthesis [14]. It includes species of *Bacillus* and related genera [20,21] called aerobic endospore-forming bacteria (AEFB). In contrast other Gram-positive cocci and *Lactobacillus* spp. exclusively rely on the MVA pathway [21]. Conversely, *Listeria* spp. and a subset of *Actinobacteria*, such as *Streptomyces* spp., utilise one or the other to produce these isoprenes. Many terpenes, such as the antimicrobial albaflavenone [22], germacrene D, and pentalenene [6], have been observed in bacteria, notably strains of *Actinomicetales* and Gram-positives. However, terpenes’ physiological and ecological roles in these organisms remain largely unknown.

The genus *Streptomyces* is considered a model for terpene studies and heterologous expression. However, recent advancements in the bioproduction of lycopene are well documented, showcasing engineered *Escherichia coli* strains bearing modified MEP pathways with the potential to develop monoterpene-enhanced wines [23]. Despite the recognisable potential for expressing BGCs, the heterologous biosynthesis of terpenes on alternative backgrounds, such as *Bacillus* species, is still scarce, lagging behind hosts such as *E. coli*, *Streptomyces* spp. and *S. cerevisiae* [21,24].

Species allocated to the genus *Bacillus* and related genera—whether assigned to the same or different orders and families—are designated aerobic endospore-forming bacteria (AEFB), and soil is considered their primary reservoir [25,26,27]. To acquire knowledge on AEFB and gain insights into their potential as sources of novel bioactive substances, including terpenes, we previously isolated 312 strains through heat-shocking soil samples collected in random areas of the Federal District, Midwest region of Brazil. These environmental strains are designated SDF0001-SDF0312 and are deposited at the Coleção de Bactérias aeróbias formadoras de endósporos (CBafes, or AEFB Collection–AEFBC). The CBafes is hosted at the University of Brasilia and is currently undergoing taxonomic classification using a polyphasic approach [28,29,30,31].

Endosporulation is an outstanding differentiation mechanism that evolved to help some bacteria survive adverse conditions [32,33]. The resulting dormant spore remains sensitive to environmental changes and can germinate to return to active metabolism and reproduction [33]. The ability to form spores has been observed only inside the phylum *Firmicutes*, recently renamed *Bacillota* [34]. This phylum allocates low G+C bacteria, most of which have Gram-positive cell wall structures, distributed in eight classes [35].

Spore formation is not a universal characteristic inside *Bacillota*, but endospore-formers share a minimal homologous gene set involved in this event [35]. Spore formation is widespread in the two *Bacillota* major classes—*Bacilli*, which are aerobic or facultative, and *Clostridia*, which are anaerobic [34,35].

Besides their remarkable spore resistance, AEFB can thrive across large temperatures and pH levels and exhibit metabolic diversity. These traits contribute to their ubiquity. These prokaryotic cells are known to synthesise a diverse array of specialised metabolites, which include terpenes [20,26,36,37,38]. Researchers extensively study these exceptional characteristics in various industrial contexts [20]. These metabolites’ ecological and socioeconomic significance is well recognised, as they can promote plant growth, manage insect pests and disease vectors, and possess immunosuppressive, antimicrobial, and antitumor activities [20,38,39,40]. Despite their potential importance, the terpene biosynthetic pathways in AEFB remain largely unexplored.

Genome prediction offers a powerful means for identifying and characterising the genetic basis of terpene production in bacteria. Here, we used a genome mining approach to investigate the potential for terpene biosynthesis in the whole genome of 10 SDF strains (Table 2). We focused on identifying the key genes involved in the MEP pathway and the enzyme of the PPS family by examining 16 BGCs we previously identified in these genomes employing the antiSMASH in silico pipeline [41]. We also sought to identify genes encoding terpene synthases (TSs), the enzymes responsible for the final steps in terpene biosynthesis. Finally, we reconstructed phylogenetic trees based on corresponding amino acid sequences of the TSs found to resemble phylogenetic relationships based on the whole genomes of the respective SDF strains. Our findings provide new insights into the diversity and evolution of terpene biosynthesis in AEFB and highlight the potential of these environmental strains as a source of novel terpenes with valuable applications.

## 2. Materials and Methods

### 2.1. Bacterial Strains

The 10 SDF strains evaluated in this study (Table 2) are deposited at the Coleção de Bactérias aeróbias formadoras de endósporos (CBafes, or Aerobic Endospore-Forming Bacteria—AEFB Collection), hosted at the University of Brasilia, Brazil. Six genomes were explicitly sequenced for this study, plus four previously sequenced genomes from the same culture collection, accessible at the NCBI. They were isolated from Brazilian soils, preserved as dry spores in filter paper, and stored at room temperature, as described in Orem et al. [28] and Cavalcante et al. [29]. It is important to note that the isolation of SDF strains and the related studies did not intend to explore ecological interactions of all kinds among soil and/or any microbial community type. Furthermore, the location for soil collection was randomly chosen.

### 2.2. Ethics Statement

The specific permissions required to collect the SDF strains used in this study were endorsed by the Federal Brazilian Authority (CNPq; Authorisation of Access and Sample of Genetic Patrimony n° 010439/2015-3). Sampling did not involve endangered or protected species.

### 2.3. Sequencing, Assembly, Annotation, and Data Availability

The total DNA of the six SDF strains sequenced explicitly for this study (Table 2) was extracted and purified using the Wizard genomic kit (Promega, Madison, WI, USA) following the manufacturer’s instructions and sequenced using an Illumina Miseq PE (San Diego, CA, CA, USA) (150 bp) platform at the Catholic University of Brasilia (Brazil). MiSeq reads were evaluated for quality control using FASTQC 0.12.0 (http://www.bioinformatics.babraham.ac.uk/projects/fastqc/, accessed on 10 May 2016), followed by trimming trichromatics (a phred quality score threshold of 33) and assembly into contigs/scaffolds using the A5-mise pipeline [42]. This pipeline automatically processes adapter trimming, quality filtering, error correction, contig, scaffold generation, and misassembly detections. The genomes were deposited at the NCBI (Table 2). Gene annotation was performed using the NCBI prokaryotic genome annotation pipeline [43].

### 2.4. Whole Genome-Based Features and Phylogeny

A whole genome-based phylogeny analysis was performed using the OrthologSorter tool [44], available at https://git.facom.ufms.br/bioinfo/orthologsorter (accessed on 25 February 2025). Orthologsorter generates, among other data, protein families shared across all genomes (core genome). Orhtologsorter employs BLASTp [45], using as parameters a BLOSUM62 substitution matrix, gap opening penalty of 11, gap extension penalty of 1, and an E-value cutoff of 1 × 10^−5^, and OrthoMCL [46] tools with default parameters (inflation value of 1.5 for the Markov clustering algorithm, a BLASTp E-value cutoff of 1 × 10^−5^, a minimum per cent match length of 50%, and a per cent identity cutoff of 30%) to determine orthology. For our set of the 10 SDF strain genomes, plus the included outgroup *Staphylococcus pseudintermedius*, 918 core families have been found. These families were aligned, and, after removing poorly aligned positions and divergent regions using GBlocks [47], the resulting whole alignment was used to build the phylogenetic tree with RAxML [48] with a PROTCATJTT substitution model, rapid bootstrapping (1000 replicates), and a subsequent maximum likelihood search.

### 2.5. BGC Predictions

The antiSMASH 6.0 bacterial standalone version [49] optimised for prokaryotic sequences (https://antismash.secondarymetabolites.org/#!/start, accessed on 4 July 2023) was run on the 10 genomes of the SDF strains to identify BGCs linked to terpene synthesis (Table 2). The accuracy parameter of the detected clusters was relaxed (full-featured run) with algorithms provided by the antiSMASH platform (KnownClusterBlast, ActiveSiteFinder, ClusterPfam, ClusterBlast, and Pfam-based GO term annotation). The BGC similarity level (0–100%) reported for a specific metabolite was obtained by crossing over data available in the Minimum Information about a Biosynthetic Gene cluster (MiBig) platform (https://mibig.secondarymetabolites.org, accessed on 1 March 2023). The percentage index indicates the number of the gene sequences within a BGC that have a hit to any gene in a particular BGC in the MiBiG’s reference strain related to terpene production.

### 2.6. MEP Pathway Reconstruction

Pathway Tools 26.5, a systems-biology software—associated with the BioCyc Pathway/Genome Database Collection (http://bioinformatics.ai.sri.com/ptools/, accessed on 1 March 2023), was used to predict the gene sequences coding for the MEP pathway catalysts (Table 1). The algorithm PathoLogic was used to create a Pathway/Genome Database (PGDB) containing the predicted metabolic pathways of the respective strains. The PGDB was built using a cutoff score of 0.15 and the inference tools Transport Inference Parser, Pathway Hole Filler, Operon Predictor, and Protein Complex Predictor in the activated mode. The Omics Dashboard tool was used to orient metabolomic data to create one diagram showing an aggregated system-oriented view of the metabolic routes of the 10 SDF strains.

### 2.7. Detection of Polyprenyl Synthase Enzymes

Data from the NCBI platform (https://ncbi.nlm.nih.gov, accessed on 1 March 2023) were used to investigate the presence of sequences coding for PPS enzymes (Table 1) in the 10 SDF strains studied (Table 2). To this end, a database containing 6146 files of amino acid sequences (.fasta) of polyprenyl synthases—obtained from species allocated to the four orders explored in this study and deposited at NCBI—was built. The extraction of protein sequences from each SDF strain genome (.gbk) was accomplished using the script available in Bogdanove et al. [50]. The alignment and comparison of the amino acid sequences between the database created and SDF genome sequences were achieved using BLASTp amino acid sequences (https://blast.ncbi.nlm.nih.gov/Blast.cgi?PAGE=Proteins, accessed on 4 July 2023) where the highest hits were considered to detect the enzyme presence in the SDF strains.

### 2.8. Similarity of the Enzyme Set for Terpene Production

The putative SDF producers, the enzymes of the MEP pathway and PPS family (Table 1), and the TSs detected for each strain were arranged in heatmaps [51] using the software R (https://www.r-project.org, accessed on 27 February 2025). The dichotomous values 0 (for the absence of catalyst) and 1 (for the presence of catalyst) were taken as binary variables representing these associations. Pearson’s correlation was employed to cluster the SDF strains, taking a similar set of enzyme results [52].

### 2.9. Phylogenetic Tree Reconstruction Based on Terpene Synthase Content

The amino acid sequences of the TSs identified were extracted from the corresponding BGC obtained by antiSMASH 6.0 [49] and using an in-house script in the Biopython programming language (http://biopython.org/DIST/docs/tutorial/Tutorial.html, accessed on 1 March 2023). Inside MEGA software version 11, the amino acid sequences (.fasta) obtained were aligned using ClustalW with default parameters (https://www.megasoftware.net/ClustalW, accessed on 18 October 2024). The file generated (.mas) was used to reconstruct phylogenetic trees employing the maximum likelihood statistical method based on 1000 bootstrap replicates.

## 3. Results

### 3.1. SDF Strain Genome Features

This study describes genomic resources for 10 cultivable environmental AEFB samples designated SDF strains (Table 2). We presented high-quality whole-genome sequences from 10 SDF strains based on Illumina. These samples corresponded to four orders, four families, six genera, and nine species allocated to the phylum *Bacillota*, class *Bacilli* (Table 2). Six samples were assigned to the order *Bacillales*, family *Bacillaceae*. Among them, four strains belonged to three different *Bacillus* spp.: *Bacillus pumilus* SDF0011, *Bacillus safensis* SDF0016, *Bacillus velezensis* SDF0141, and *Bacillus velezensis* SDF0150. The family *Bacillaceae* was also represented by two other genera and species of named strains, *Heyndrickxia oleronia* SDF0015 and *Peribacillus simplex* SDF0024, referred to here as *Pe. simplex* SDF0024. Inside the order *Caryophanales*, the family *Caryophanaceae*, genus *Lysinibacillus* were represented by two strains, *Lysinibacillus fusiformis* SDF0005 and *Lysinibacillus sphaericus* SDF0037. The strain *Paenibacillus popilliae* SDF0028 belonged to the order *Paenibacillales*, family *Paenibacillaceae*, and genus *Paenibacillus*. Finally, *Brevibacillus brevis* SDF0063 was allocated to the order *Brevibacillales*, family *Brevibacillaceae* and is referred to here as *Br. brevis* SDF0063. Genome analysis of the SDF strains uncovered considerable differences in genome size, scaffold number, N50, GC content, coding sequences (CDSs), protein-coding regions, pseudo genes, rRNAs, and tRNAs, as detailed in Table 2. Briefly, the genome sizes ranged from 3,674,191 to 6,580,875 bp, with the scaffold numbers varying from 15 up to 75 and GC content (%) spanning from 34.7 to 47.3.

A maximum likelihood method was applied to reconstruct a phylogenetic tree based on the results of the OrthologSorter tool (available in https://git.facom.ufms.br/bioinfo/orthologsorter, accessed on 25 February 2025) [44] in 10 SDF strain whole-genomes (Table 2) and *S. pseudintermedius* as an outgroup (Figure 1), which resulted in two major clades. The *Lysinibacillus* spp., *L. fusiformis* SDF0005, and *L. sphaericus* SDF0037 clustered together in the most distinct branch. The genomes of *H. oleronia* SDF0015 and *Pe. simplex* SDF0024 formed a branch that also included the four *Bacillus* strains, *B. pumilus* SDF0011, *B. safensis* SDF0016, *Bacillus velezensis* SDF0141, and *Bacillus velezensis* SDF0150, on the second major clade. The strains *P. popilliae* SDF0028 and *Br. brevis* SDF0063 were also positioned as the most distinct SDF strains analysed.

### 3.2. MEP Pathway Reconstruction

BGCs are a locally clustered group of two or more genes in a particular genome. antiSMASH is an in silico pipeline offering the detection and analysis of many BGC types [49]. These gene clusters encode biosynthetic pathways for specialised metabolite production with diverse functions, including chemical variants [53]. Previously, using the antiSMASH 6.0 bacterial standalone version [49], we identified 153 putative BGCs codifying for 20 different classes of specialised metabolites synthesis in 10 SDF strains (Table 2) deposited at CBafes [41]. Among these, 16 were related to terpene synthesis. In this work, the potential of these SDF strains for terpene biosynthesis was further addressed by taking advantage of these 16 high-quality BGC sequences.

The algorithm PathoLogic (http://bioinformatics.ai.sri.com/ptools/, accessed on 1 March 2023) predicted that all the corresponding gene sequences coding for the seven enzymes (DXS, DXR, MCT, CMK, MDS, HDS, and HDR) that catalyse the MEP pathway reactions (Table 1) would be found among the 10 SDF genomes (Table 2). The PGDB obtained is represented in a diagram aggregating a system-oriented view of the metabolic routes of the 10 SDF strains generated by the Omics Dashboard tool (Figure 2). In addition to MEP route enzymes, this tool also detected the enzyme IDI (Table 1), responsible for both IPP isomerisation to DMAPP and the subsequent IPP and DMAPP condensation that generates the first substrate in the terpenes’ production (Table 1). The information coding for the enzyme IDI was present in all SDF strains analysed, except for the *L. fusiformis* SDF0005 strain (Figure 2).

### 3.3. Detection of Polyprenyl Synthase Enzymes

Using BLASTp, the alignment and comparison of the amino acid sequences between the database we created (6146 files; see Section 2) and SDF sequences revealed the presence of the PPS family —the enzymes responsible for the conversion of IPP to GPP, FPP, and GGPP (Table 1). The strains *L. fusiformis* SDF0005; *H. oleronia* SDF0015; *Pe. simplex* SDF0024; *B. velezensis* SDF0141; and *B. velezensis* SDF0150 presented a >98% amino acid similarity, while *H. oleronia* SDF0015 exhibited a 67.86% one (Table 3). The amino acid sequences for PPS were not detected in the remaining SDF strains.

### 3.4. Prediction of Biosynthetic Gene Clusters Associated with Terpene Synthesis

At least three gene sequences inside the 16 BGCs uncovered by antiSMASH [41] directed the synthesis of three TSs: (i) a *sqhC* gene, coding for a squalene-hopene cyclase (SHC); (ii) a gene encoding undetermined activity related to the production of the phytoene and/or squalene synthase (PSS); and (iii) a *crti* gene, coding for a phytoene desaturase (PDS). Table 3 outlines the TS coding sequences for the strains used in this study.

Out of these 10 SDF strains, the sequences coding for the SHC were detected in eight genomes: *B. pumilus* SDF0011; *H. oleronia* SDF0015; *B. safensis* SDF0016; *Pe. simplex* SDF0024; *P. popilliae* SDF0028; *Br. brevis* SDF0063; *B. velezensis* SDF0141, and *B. velezensis* SDF0150 (Table 4; Figure 3). The sequences coding for the enzymes of the PSS synthase family were found in seven genomes: *L. fusiformis* SDF0005, *B. pumilus* SDF0011, *B. safensis* SDF0016, *Pe. simplex* SDF0024, *L. sphaericus* SDF0037, *B. velezensis* SDF0141, and *B. velezensis* SDF0150 (Table 4; Figure 4). The BGC structures involved in terpene production varied among the analysed strains, as detected by antiSMASH. Depending on the species, the genes responsible for synthesising SHC and PSS were flanked by different genetic elements within the BGCs. However, an exception was noted for the strains of *B. velezensis* SDF0141 and *B. velezensis* SDF0150, which exhibited structurally similar BGCs (Figure 3 and Figure 4).

In contrast, the genomes of *B. pumilus* SDF0011 and *Bacillus safensis* SDF0016 contained the genes encoding the PSS and SHC enzymes positioned adjacent to two copies of the *crtI* gene, which codes for a PDS (Figure 4). The BGC containing these two gene copies in the genomes of these two strains are reported to take part in carotenoid production [54]. Furthermore, these latter BGCs presented a 50% similarity compared to the corresponding sequence described for strain *Halobacillus halophylus* DSM2266 in the MIBiG platform (reference number BGC0000645) used by antiSMASH as a reference to predict this metabolite. The similarity percentages (0–100%) indicate the number of genes within the reference that have a hit to any genes in a particular BGC related to terpene production and that were recognised by antiSMASH.

### 3.5. Distribution of the Enzyme Set for Terpene Production Among the 10 SDF Strains

The distribution of the predicted enzymes obtained from the in silico translation of the corresponding gene sequences coding for the seven MEP pathway enzymes (DXS, DXR, MCT, CMK, MDS, HDS, and HDR), along with the enzyme IDI, were also analysed in the SDF strain genomes. The PPS family, the three TS enzymes SHC, PSS, and PDS, were also included. To this end, Pearson’s correlation [52] was employed to cluster the 10 SDF strains (Table 2; Figure 5) based on the ensemble of enzymes engaged in the terpene’s synthesis detected. We constructed a heatmap [51] representing the presence or absence of an enzyme in a particular strain to enhance the potential visual distribution of the enzyme distributions among the strains. Two major clades were distinguished (Figure 5). The first comprised *B. pumilus* SDF0011 and *B. safensis* SDF0016, which shared an identical profile, or 11 out of the 12 enzymes detected. The second major clade was further split. *Pe. simplex* SDF0024, *B. velezensis* SDF0141, and *B. velezensis* SDF0150 also shared the same profile, bearing an equivalent set of 11 enzymes (Figure 5). The next subclade embraced *Lysinibacillus sphaericus* SDF0037 and *L. fusiformis* SDF0005, despite the dissimilar enzymatic profiles (Figure 5). Although *H. oleronia* SDF0015 is the only representative of a subclade, the correlation showed that the enzymatic set for terpene production in this strain was compatible with the subclade encompassed by *P. popilliae* SDF0028 and *Br. brevis* SDF0063, except that one out of nine enzymes was missing.

### 3.6. SDF Strains’ Evolutionary Relationship Based on Two TS Amino Acid Sequences

As described above, the BGC sequences involved in the terpenes’ production detected by the antiSMASH tool encompassed three TS enzyme gene sequences (Table 4). A phylogenetic tree was reconstructed based on multiple alignments of the eight SHC amino acid sequences obtained by in silico translation from the corresponding gene sequence of these SDF strains (Figure 6A). The amino acid sequence of *Pseudomonas* sp. was included as an outgroup. The inferred evolutionary relationship among these SDF strains was clustered into two main clades. In the first, the respective SHC sequences found in the strains *B. velezensis* SDF0141 and *B. velezensis* SDF0150 were equivalent. The enzyme sequence obtained for *B. pumilus* SDF0011 was very close to that obtained for *B. safensis* SDF0016. The SHC sequences from *H. oleronia* SDF0015 and *Pe. simplex* SDF0024 revealed the highest evolutionary relationship in the second main clade generated (Figure 6A). The molecular correlation between the SHC primary chains of these two strains was closer to the corresponding sequences obtained for the strains *P. popilliae* SDF0028 and *Br. brevis* SDF0063, also positioned in the second clade (Figure 6A).

Likewise, the phylogenetic tree generated from seven amino acid sequences corresponding to the PSS enzyme family genomes and *Pseudovibrio brasiliensis* as an outgroup also divided these seven SDF strains into two main clades (Figure 6B). The first was further separated into two subclades. The sequences of this enzyme obtained for the strains *B. velezensis* SDF0141 and *B. velezensis* SDF0150 pointed out a close evolutionary relationship. The sequences of the strains *B. pumilus* SDF0011 and *B. safensis* SDF0016 positioned in the other subclade also displayed a high-level molecular relationship (Figure 6B). The second clade showed that the strains *L. fusiformis* SDF0005 and *L. sphaericus* SDF0037 presented amino acid sequences with the highest molecular relationship. Strain *Pe. simplex* SDF0024 was positioned apart from the other two inside this clade (Figure 6B). Because the gene sequence of PDS was found in 2 out of 10 SDF strains, it was not considered for further phylogenetic analyses.

## 4. Discussion

The phylum Firmicutes was recently renamed *Bacillota* [34]. Inside the class *Bacilli*, the order *Bacillales*, which allocated AEFB species, displayed an immense diversity, spanning several families, genera, and species [20,35]. Lately, new taxa have been established to reposition strains otherwise considered members of the order *Bacillales* (https://lpsn.dsmz.de/contact, accessed on 19 December 2024) [55]. This reorganisation considers the family *Bacillaceae* the only member of this order, includes *Bacillus* as the type genus of these taxa, and *Bacillus subtilis* remains the type species of the genus *Bacillus*.

Historically, the genus *Bacillus* represented a large assemblage of genetically and evolutionarily unrelated microorganisms. Thus, the genus has long been recognised as housing members exhibiting an extensive polyphyly and with very little in common with each other [34,55,56,57]. To more adequately represent the overall genetic diversity within this genus, it was proposed that the vast majority of *Bacillus* spp. needed to be reclassified into other genera, families, and orders. The revision of the genus *Bacillus* led to reallocating (not limited to) two *Bacillus* spp. to novel genera that could better accommodate them. The former *Bacillus oleronius* is now designated *Heyndrickxia oleronia* [58]. Likewise, *Bacillus simplex* was reallocated into a novel genus designated *Peribacillus*, species *Peribacillus simplex* [59].

Other misclassified *Bacillus* spp. were transferred to specific genera and reallocated to the family *Caryophanaceae*, which in turn were transferred to the order *Caryophanales*, including the genus *Lysinibacillus* [56]. In this context, the genera *Paenibacillus* and *Brevibacillus* were moved from the genus *Bacillus*, family *Bacillaceae*, being the genus *Paenibacillus,* to the family *Paenibacillaceae*, order *Paenibacillales*, and the genus *Brevibacillus* to the novel family *Brevibacillaceae*, order *Brevibacillales* [57].

### 4.1. Uncovering Enzymes from the MEP Pathway and the Polyprenyl Synthase Family in the SDF Strains

We previously identified 153 putative BGCs in the genomes of 10 SDF strains (Table 2). Among these, 20 classes of specialised metabolites were identified [41]. In the current work, we focused on 16 BGCs, which were linked to terpene synthesis, to assess the genomic potential of these environmental AEFB for the biosynthesis of these molecules. Bacteria can produce terpenes through the MEP pathway by synthesising IPP, a precursor for these isoprenes [14]. Therefore, we evaluated the putative genetic information associated with the biosynthesis of essential terpene precursors within the MEP pathway. Through the Pathways tools, our study has uncovered that all the 10 SDF strains (Table 2) examined contain a significant number of genetic determinants encoding the enzymes DXS, DXR, MCT, CMK, MDS, HDS, and HDR, which are associated with the MEP pathway (Table 1; Figure 2). These findings suggest that the genetic information coding for these pathway enzymes was conserved among these SDF strains, which can be carriers of the basic apparatus for terpene production.

Indeed, information for MEP pathway catalysts (Table 1) is an ancestral characteristic of prokaryotes. According to Zeng and Dehesh [60], there is substantial evidence of the vertical transfer of genetic information encoding MEP pathway constituents between plastids—initially present in the phylum *Cyanobacteria*—and plants. This evidence supports the idea that the genetic determinants of the MEP pathway are an ancestral characteristic shared among the SDF strains evaluated in this study and potentially also by AEFB. For instance, the genetic information related to enzyme production found in the *B. velezensis* SDF0150 strain has also been identified in the genome of *Synechocystis* sp., which is classified under the phylum *Cyanobacteria* [61]. Additionally, this metabolic pathway is well characterised in the phylum *Actinomycetota*, notably in species of the genus *Streptomyces* [62]. These compelling data underscore the necessity for further investigation to explore the potential implications of these fundamental genetic determinants in terpene biosynthesis within AEFB.

The PPS family amino acid sequences obtained by in silico translation (Table 3) were compared to a sequence database we assembled, with more than 6000 trusted files of known sequences involved in these catalysts’ synthesis (see Section 2). We performed pairwise all-against-all protein sequence alignments of all the genomes, using Blastp with an E-value cutoff of 1 × 10^−5^. The best hits were found to be *B. velezensis* SDF0141 and *B. velezensis* SDF0150, in which the identity scored >99% of the top-scoring hit in the database (Table 3), an ortholog of a *B. velezensis* strain sequence. Likewise, *L. fusiformis* SDF0005 and *Pe. simplex* SDF0024 scored a >98% identity (Table 3) if compared to the top-scoring hit in the database representative of their respective species. These findings strongly indicate the presence of the gene sequences coding for this enzyme in the genome of these SDF strains.

In contrast, *H. oleronia* SDF0015 showed a 67.86% identity if compared to the ortholog sequences representing *B. pumilus* in the database. (Table 3) This result suggested that this latter strain may lack the genetic information necessary for synthesising the PPS family enzyme. Alternatively, it possesses a structurally distinct catalyst compared to those found in AEFB, likely due to unique characteristics associated with *H. oleronia*. Consequently, it is reasonable to infer that the remaining SDF strains lacking at least one enzyme from the PPS family may experience limitations in terpene production, as they do not bear the necessary enzymatic machinery to synthesise intermediates for isoprene precursors.

### 4.2. Genomic Potential of Selected SDF Strains for Terpene Production

GPP, FPP, or GGPP are used as precursor molecules in natural terpene syntheses by different TSs. antiSMASH identified the sequence encoding the enzyme of the PSS in at least seven SDF strains (Figure 4; Table 4). Phytoene and squalene molecules are constructed from two molecules of FPP and two GGPP molecules, respectively [63,64]. The enzymes squalene synthase (SQS) and phytoene synthase (PHS) are closely related [64], and SQS is also reported for the synthesis of phytoene [63]. This connection between these enzymes may explain why antiSMASH did not discriminate between the genetic sequences for phytoene and squalene production. The SDF strains containing genes that encode enzymes from the PPS family and PSS putatively presented the necessary genetic apparatus to produce either phytoene or squalene, as illustrated in the genomes of *Pe. simplex* SDF0024, *B. velezensis* SDF0141, and *B. velezensis* SDF0150.

Squalene is known for its antioxidant and antitumor properties, and this terpene also enhances the human immune system, as reported by Sanchez-Quesada et al. [65]. This specialised metabolite is commonly used as an additive and supplement in the food and personal care industries [66]. Squalene synthesis has been documented for various organisms, including AEFB, which corroborates the findings in this study. Beyond its benefits, squalene serves as an intermediate compound in the biosynthetic pathway of sterols such as hopanoids [67].

Hopanoids are synthesised from squalene by the enzyme SHC [67]. The *sqhC* gene sequence encodes the SHC enzyme and was identified by antiSMASH in eight SDF strains (Figure 3; Table 4). These data indicate that the SDF strains qualified to produce SHC and PSS possess the required components for the final enzymatic reactions in hopanoid biosynthesis. This condition was observed in three SDF genomes, specifically *Pe. simplex* SDF0024, *B. velezensis* SDF0141, and *B. velezensis* SDF0150 (Table 4). Hopanoids play a crucial role by integrating into the biological membranes of the producing cells, regulating fluidity and permeability [67,68]. Consequently, these terpenes keep the bacterial cytoplasmic membrane stable, which is particularly significant given the absence of cholesterol in the membranes of these prokaryotes. While a lack of hopanoids does not hinder bacterial growth, it does affect tolerance to stressful conditions, such as high temperatures and anaerobic or acidic environments [69].

Interestingly, the antiSMASH analysis revealed that the organisation of hopanoid biosynthesis genes in the SDF strains deviated from the typical BGC pattern. While the *sqhC* gene encoding SHC was identified in eight strains, and genes encoding PSS enzymes were found in seven, these genes were not consistently clustered with other hopanoid biosynthesis genes (Figure 3 and Figure 4). This variability suggests that the genetic architecture of hopanoid biosynthesis in SDF strains might be more complex and diverse than previously recognised. Further investigation into the arrangement and regulation of these genes may shed light on this variation in its evolutionary and functional significance. Despite this variability, key hopanoid biosynthesis gene identifications in the genomes of these strains underscore the potential of AEFB as a source of these important membrane components.

The software antiSMASH 6.0 bacterial standalone version [49] (https://antismash.secondarymetabolites.org/#!/start, accessed on 4 July 2023) detected a 50% similarity in the BGC sequence involved in the carotenoid’s synthesis in the genomes of *B. pumilus* SDF0011 and *B. safensis* SDF0016. The reference strain employed for comparison was *Halobacillus halophilus* DSM2266 (reference number BGC0000645), available through the MIBiG platform. Although the SDF strains did not share the same genetic framework as the reference strain, they possessed the essential genetic information required to express the final enzymatic activities involved in lycopene synthesis. Lycopene, which closely resembles beta-carotene and is widely produced by plants [70], imparts the reddish pigmentation characteristic of tomatoes and watermelons, among other vegetables [71]. This terpene is known for its antioxidant, anti-inflammatory, and antitumor attributes, making it valuable for various applications in the pharmaceutical and food industries.

It is noteworthy that lycopene synthesis in AEFB has been reported in the context of heterologous expression in *B. amyloliquefaciens* and *B. subtilis*, as documented by Zou et al. [72] and Luo, Bao, and Zhu [73]. Additionally, studies have explored natural lycopene production in other AEFB species. For instance, Osawa et al. [74] investigated the synthesis of an oxidised lycopene in *Cytobacillus firmus* [59], formerly *Bacillus firmus* [75], while Hwang et al. [76] also detected genes for lycopene synthesis in *Metabacillus flavus*. The findings of our study are consistent with these previous reports and underscore the potential of AEFB as a source of lycopene. Further investigation is needed to fully elucidate these mechanisms and the evolutionary significance of lycopene production in these bacteria.

Figure 7 summarises the catalytic steps required for terpene production from the MEP pathway to the final reactions by different TSs and the SDF strain carriers for their respective enzymes. Nonetheless, it is essential to note that even though a specific strain of SDF may lack a particular enzyme required for terpene synthesis, this does not automatically imply that the cell is a non-terpene producer. Our study concentrated on the genomic potential of the SDF strain to generate terpenes. Therefore, any inaccuracies in the prior steps, such as the purification, extraction, sequencing, and annotation of genome sequences, may lead to erroneous results. Additionally, the biosynthetic pathways of terpenes entail multiple enzymatic reactions [77]. Even if the information for a specific enzyme is not detected, the SDF strains examined might still be able to synthesise the predicted terpene because of the promiscuous nature of the enzymes found within the pathway. A complete enzymatic route detection for a given terpene molecule does not assure the synthesis of the corresponding product by this SDF strain due to the complex mechanisms of gene expression. Further research could reveal the synthesis in vitro of the molecules detected in this study.

### 4.3. The Evolutionary Nature of Terpene Production in the SDF Stains

To explore the evolutionary nature of the enzymes involved in terpene production in the evaluated SDF strains, we aligned the amino acid sequences of the SHC (Figure 3) and PSS enzymes (Figure 4) obtained through in silico translation from the gene sequences inside the BGC identified by antiSMASH in this study. This alignment was used to generate phylogenetic trees, as shown in Figure 6A,B. However, we did not reconstruct the phylogenetic tree for the PDS enzyme, as it was detected only in two SDF strains analysed (Table 4).

The primary sequences of the SHC (Figure 6A) and PSS (Figure 6B) enzymes of *B. velezensis* SDF0140 and *B. velezensis* SDF0150 exhibited a robust molecular relationship, as both strains belong to the same species. In contrast, the strains *B. pumilus* SDF0011 and *B. safensis* SDF0016 showed a significant evolutionary relationship based on their TS amino acid sequences. These two latter species are of biotechnological and pharmaceutical significance and are closely related according to classical phenotypic characteristics and *16S rRNA* gene sequences. Consequently, they are challenging to distinguish by these conventional methodologies [28,30,31,78]. Both species comprise a clonally diverse population inside the *B. subtilis* complex [78].

Furthermore, the phylogenetic trees generated using both the SHC (Figure 6A) and PSS (Figure 6B) enzymes revealed similar topologies among *B. pumilus* SDF0011, *B. safensis* SDF0016, *B. velezensis* SDF0141, and *B. velezensis* SDF0150. This result indicates that these four *Bacillus* spp. demonstrate a high level of molecular correlation in their respective TS analyses. Their classification within the same genus likely contributes to this significant conservation of catalytic properties.

Interestingly, the molecular relationship observed in Figure 6B shows that the PSS enzyme positions the two species of the genus *Lysinibacillus* (*L. fusiformis* SDF0005 and *L. sphaericus* SDF0037) and *P. simplex* SDF0024 in the same clade. This grouping contrasts with the global genome phylogeny described in Figure 1. Thus, it may suggest a possible horizontal transfer of this sequence at some point in the evolution of these taxa allocated to different orders of the class *Bacilli*. However, further investigation is necessary since the molecular relationships for the other SDF strains align with the expected phylogenetic relationships (Figure 6A,B).

In the heatmap (Figure 5), the enzymatic profiles for terpene production in the SDF strains highlighted a strong molecular relationship between *B. pumilus* SDF0011 and *B. safensis* SDF0016, as they share the same enzymatic set. Figure 5 also reveals that *B. velezensis* SDF0141 and *B. velezensis* SDF0150 possess an identical catalyst set for terpene production. These results reinforce the molecular relationship among these SDF strains, although the disposition of SDF strains belonged to *Bacillus* spp. did not display the same distribution observed in the phylogenetic tree of TS amino acid sequences (Figure 6).

The strains *H. oleronia* SDF0015 and *Pe. simplex* SDF0024 exhibited a high degree of molecular similarity in their respective amino acid sequences of the SHC enzyme (Figure 6A). The amino acid sequences of SHC from *H. oleronia* SDF0015 and *Pe. simplex* SDF0024 displayed a superior evolutionary distance compared to SHC sequences obtained from the SDF strains within the genus *Bacillus*, therefore positioned in different clades (Figure 6A). *Bacillus*, *Heyndrickxia*, and *Peribacillus* are all classified within the family *Bacillaceae,* and species in these genera present significant levels of polyphyly [56]. This observation further corroborated phylogenetic relationships derived from the SHC amino acid sequences (Figure 6A).

*P. popilliae* SDF0028 and *Br. brevis* SDF0063 presented the highest molecular relationship in their SHC amino acid sequences if compared to the SHC sequences of the remaining six SDF strains analysed (Figure 6A). The genera *Bacillus*, *Peribacillus*, and *Heyndrickxia*, part of the family *Bacillaceae*, belong to the order *Bacillales*. At the same time, the genera *Paenibacillus* and *Brevibacillus* are classified under the orders *Paenibacillales* and *Brevibacillales*, respectively [57,58]. Therefore, the minimal molecular relationship between the SHC amino acid sequences of *P. popilliae* SDF0028 and *Br. brevis* SDF0063 and the other SDF strains aligns with the anticipated evolutionary distance for these species. Additionally, the same enzymatic set for terpene production was observed between these two SDF strains (Figure 5), supporting even more the phylogenetic relationship obtained by the sequence of the SHC of *P. popilliae* SDF0028 and *Br. brevis* SDF0063 (Figure 6A). Notably, the amino acid sequences of the SHC from *H. oleronia* SDF0015 and *Pe. simplex* SDF0024 were recognised to be evolutionarily closer to those of *P. popilliae* SDF0028 and *Br. brevis* SDF0063 than the SHC sequences from SDF strains within the genus *Bacillus* (Figure 6A).

*L. fusiformis* SDF0005 and *L. sphaericus* SDF0037 shared highly conserved PSS amino acid sequences (Figure 6B). This commonality might be attributed to the two SDF strains from the genus *Lysinibacillus*. The phylogenetic tree generated using the sequences revealed a clear distinction between the SDF strains of the genus *Bacillus*, which were grouped in one clade, and the SDF strains of the genus *Lysinibacillus*, which formed a separate clade (Figure 6B). These results discriminated between the members of the families *Bacillaceae* and *Caryophanaceae* based on the amino acid sequence of this catalyst. Furthermore, a relevant molecular relationship for the terpene production enzymatic set was already observed between *Lysinibacillus* spp. as shown in the heatmap (Figure 5). Markedly, the sequences of *L. fusiformis* SDF0005 and *L. sphaericus* SDF0037 demonstrated a closer evolutionary relationship with the sequence of *Pe. simplex* SDF0024, which is allocated within the family *Bacillaceae*.

The phylogenetic trees derived from the molecular analyses of the TSs detected in the SDF strains examined in this study agreed with the phylogenetic relationships for the complete genomes of the SDF strains evaluated (Figure 1). The results indicated that the TS enzymes responsible for terpene production in these investigated environmental strains are evolutionarily conserved. In addition, the production of terpenes—strikingly, squalene and hopanoids—appears to be an ancestral characteristic of the AEFB evaluated in this study.

As mentioned above, the phylogenetic trees derived from the amino acid sequences of TS enzymes (Figure 6A,B) indicated a closer molecular relationship between *B. velezensis* SDF0141 and *B. velezensis* SDF0150 to *B. pumilus* SDF0011 and *B. safensis* SDF0016. In contrast, the heatmap (Figure 5) groups these two *B. velezensis* strains with *Pe. simplex* SDF0024, along with *L. fusiformis* SDF0005 and *L. sphaericus* SDF0037. Additionally, the strains *P. popilliae* SDF0028 and *Br. brevis* SDF0063 demonstrated a similar enzyme content for terpene production and clustered with the strain *H. oleronia* SDF0015 (Figure 5).

The cluster of catalysts implicated in terpene production shown in Figure 5 shares similarities with the phylogenetic trees generated from the TS amino acid sequences (Figure 6A,B). Nevertheless, the cluster does not fully align with the expected evolutionary relationships among these species. This analysis suggested that the ability to produce terpene molecules may vary among the SDF strains and might not be influenced by phylogenetic factors. Furthermore, the enzyme set involved in terpene production cannot be used as a molecular marker to establish evolutionary relationships among the SDF strains analysed and, by extension, for AEFB.

This study is based on in silico analyses. Thus, experimental approaches to confirm terpene production potential, alongside possible active applications, are significant. In future work, we plan to include transcriptomic analyses under conditions that induce a specialised metabolism. This approach will allow us to verify the expression of genes associated with the predicted terpene biosynthetic pathways. Techniques such as RNA sequencing (RNA-Seq) and quantitative-reverse transcription polymerase chain reaction (RT-qPCR) will allow us to assess whether these genes are transcriptionally active, and the conditions in which they are expressed. For instance, RT-qPCR permits the precise quantification of a specific gene expression by converting mRNA into cDNA, followed by the real-time monitoring of amplification using fluorescent markers. Still, transcriptomic data can be complemented with metabolomic profiling using mass spectrometry to detect and quantify terpene compounds produced by these environmental strains. This integrative approach will allow us to correlate gene expression with metabolite production, providing strong evidence for the functional activity of the predicted pathways.

## 5. Conclusions

AEFB are ubiquitous and characterised by producing several specialised metabolites, among which terpenes are a significant class. While the synthesis of terpenes has been demonstrated in prokaryotes, research addressing specifically the production of these compounds in AEFB is still scarce. Our analyses revealed that strains of these taxa possess the BGCs required to synthesise at least three terpenes: squalene, hopanoids, and lycopene. We successfully identified the metabolic pathways for synthesising terpene precursors in all 10 genomes of the evaluated SDF strains. The detected amino acid sequences of terpene synthases indicated functional equivalence among these catalysts. Although the identified terpene classes represent only a narrow fraction of these isoprene molecules, our findings can support future investigations to broaden the understanding of the physiological and ecological roles of terpenes in AEFB. Natural sources for terpenes are insufficient to meet the growing need, since these metabolites are extensively used across numerous industries. Therefore, synthetic biology and metabolic engineering methods can be used to create cell factories for the enhanced production of terpenes. These approaches grant the study and manipulation of BGCs under controlled parameters. Besides leading to novel functions, these strategies can help minimise the TS enzyme’s promiscuity by adding precursors that can directly synthesise a particular product and reduce the metabolic burden.

## Figures and Tables

**Figure 1 microorganisms-13-02528-f001:**
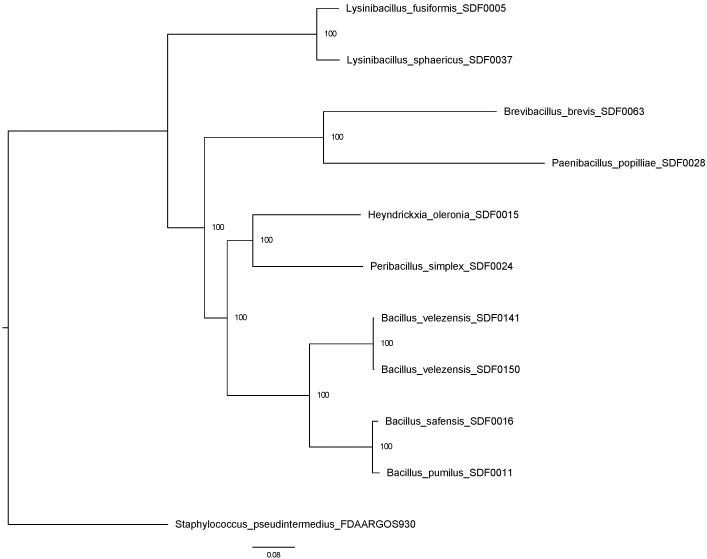
Phylogenetic relationship of the ten SDF strains based on a whole-genome analysis. An unrooted maximum likelihood tree was constructed using RAxML (PROTCATJTT model) with 918 core protein families. Bootstrap values (1000 replicates) are shown at the nodes. *Staphylococcus pseudintermedius* was used as an outgroup. Strain classifications are indicated in the branches, and a distance scale bar is displayed at the bottom.

**Figure 2 microorganisms-13-02528-f002:**
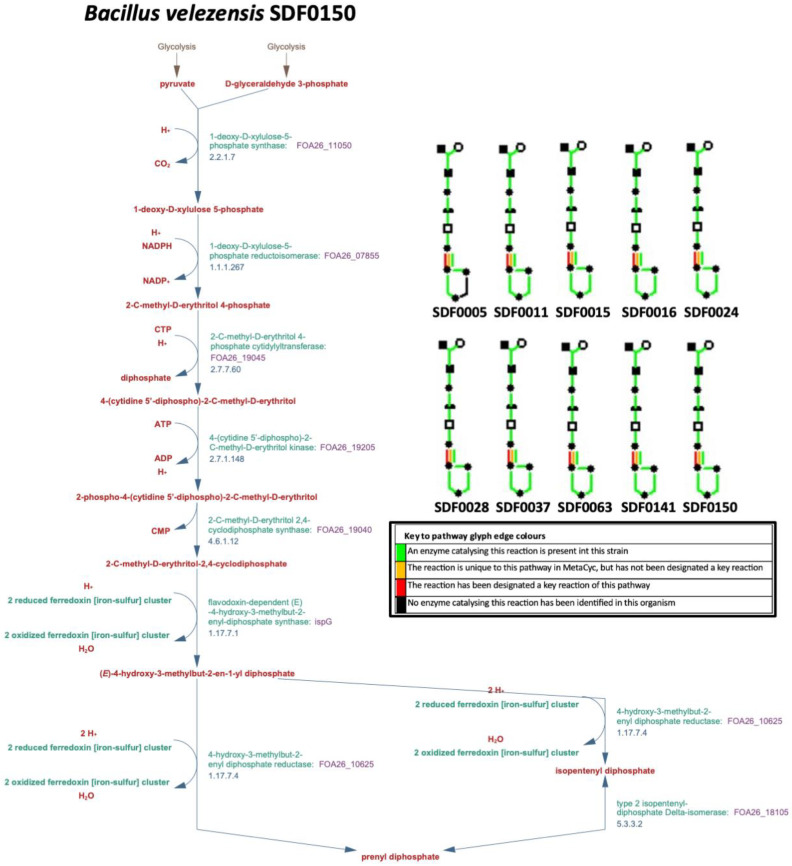
Methylerythrotol-phosphate pathway reconstruction. The Pathway Tools software created a Pathway/Genome Database (PGDB) that includes the predicted metabolic pathways of the respective strains. The Omics Dashboard tool was then applied to align the metabolomic data, generating diagrams to provide an aggregated system-oriented view of the predicted metabolic route information found in the genome of *Bacillus velezensis* SDF0150 (larger diagram). Smaller diagrams representing these routes in *Bacillus velezensis* SDF0150 and the remaining nine strains are displayed in the top right corner. A key to the colours of the pathway glyph edges is indicated. The SDF strain designation is indicated: *Lysinibacillus fusiformis* SDF0005, *Bacillus pumilus* SDF0011, *Heyndrickxia oleronia* SDF0015, *Bacillus safensis* SDF0016, *Peribacillus simplex* SDF0024, *Paenibacillus popilliae* SDF0028, *Lysinibacillus sphaericus* SDF0037, *Brevibacillus brevis* SDF0063, and *Bacillus velezensis* SDF0141.

**Figure 3 microorganisms-13-02528-f003:**
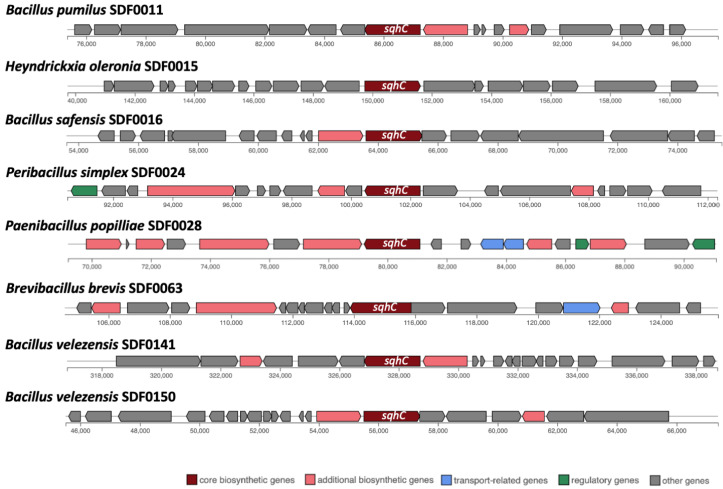
Structure of biosynthetic gene clusters involved in squalene-hopene cyclase expression in the genome of the SDF strains. AntiSMASH identified *sqhC*, which directs the production of the squalene-hopene cyclase (SHC) as a core biosynthetic gene (brown) in eight strains. Predicted gene functions (colour-coded) are shown at the bottom.

**Figure 4 microorganisms-13-02528-f004:**
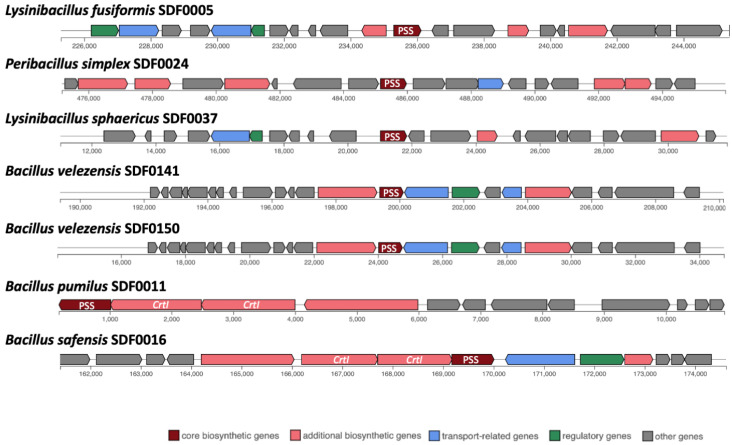
Biosynthetic gene clusters involved in the phytoene/squalene synthase and phytoene desaturase in SDF strains. A gene sequence that directs the phytoene and/or squalene synthase (PSS) family enzyme production was identified as a core biosynthetic gene (brown) inside the BGCs detected in seven SDF genomes by antiSMASH. In *Bacillus pumilus* SDF0011 and *Bacillus safensis* SDF0016, the gene codifying PSS was adjacent to a copy of the gene *crtI*, which drives phytoene desaturase synthesis. The colour boxes (bottom) indicate the predicted gene functions in the biosynthesis of terpenes.

**Figure 5 microorganisms-13-02528-f005:**
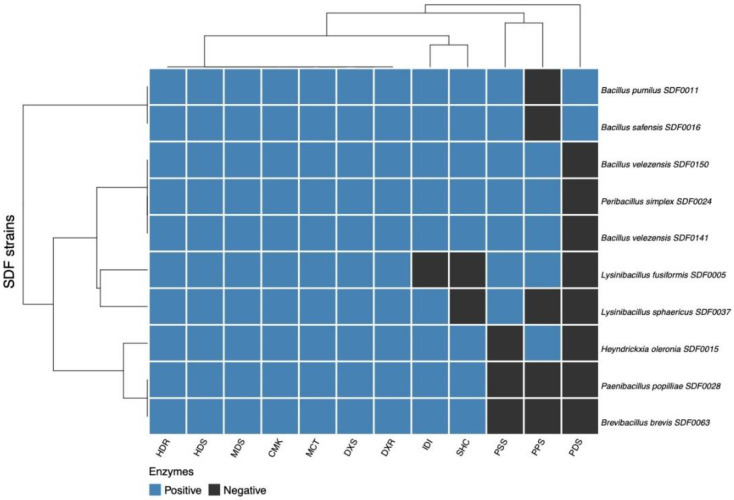
Distribution of the ensemble of enzymes engaged in the terpenes’ synthesis among the ten SDF strains. A heatmap (Person’s correlation-based) shows the SDF strains clustered into two sections based on the presence (blue squares) or absence (black squares) of the respective gene coding for the enzyme in a particular species genome (right). The protein set (Table 1) includes catalysts of the MEP route detected by the Pathways tools, along with the polyprenyl synthase family (PPS) detected by BLASTp, and the TS squalene-hopene cyclase, phytoene and/or squalene synthase (PSS), and phytoene desaturase (PDS) identified by antiSMASH are described on the bottom.

**Figure 6 microorganisms-13-02528-f006:**
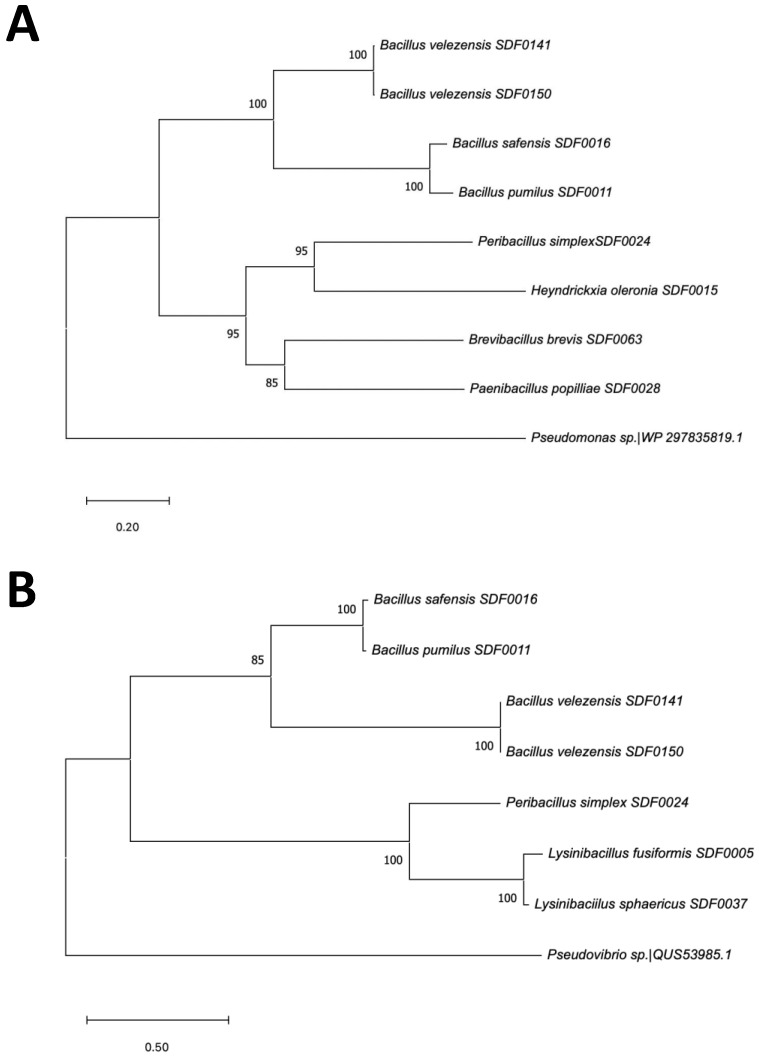
Phylogenetic relationship of the SDF strains based on terpene synthase sequences. The evolutionary history of the SDF strains was determined by aligning the deduced amino acid sequences of two terpene synthases using ClustalW. The sequences were obtained by translating the respective SDF gene sequences in silico. Phylogenetic trees were reconstructed using the maximum likelihood method in MEGA version 11.0. The tree nodes show bootstrap values as percentages of 1000 replications. SDF strain designations are indicated in the branches. A distance scale bar is displayed at the bottom. The evolutionary relationships among (**A**) eight SDF strains based on the amino acid sequence of the SHC enzyme and the homolog amino acid sequence of *Pseudomonas* sp. as an outgroup and (**B**) seven SDF strains based on the amino acid sequence of the phytoene/squalene synthase family (PSS) and the homolog amino acid sequence of *Pseudovibrio brasiliensis* as an outgroup.

**Figure 7 microorganisms-13-02528-f007:**
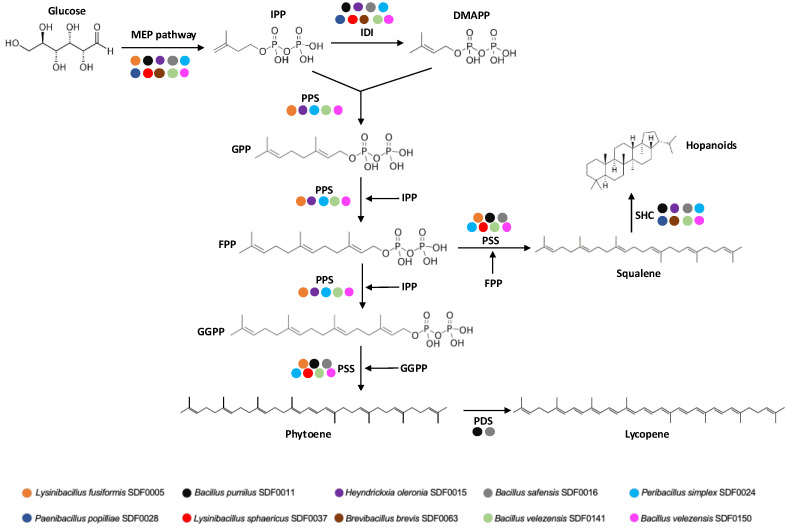
Putative biosynthetic pathway of methylerythritol phosphate and terpene biosynthesis metabolic pathways in the ten SDF strains studied. The reaction steps to synthesise terpenes from isopentenyl diphosphate (IPP), the final product of the methylerythritol-phosphate (MEP) route, and the respective catalysts are indicated. The coloured dots indicate the gene sequence coding for an enzyme detected in a particular SDF strain described at the bottom. DMAPP: dimethylallyl pyrophosphate. IDI: isopentenyl diphosphate isomerase. GPP: geranyl pyrophosphate. FPP: farnesyl pyrophosphate. GGPP: geranyl-geranyl pyrophosphate. PSS: phytoene and/or squalene synthase. SHC: squalene-hopene cyclase. PDS: phytoene desaturase.

**Table 1 microorganisms-13-02528-t001:** Enzymes involved in terpene biosynthesis. Profile of MEP pathway and polyprenyl synthase family molecules.

Substrate	Enzyme Code *	Enzyme Name (Abbreviation)	Product (Abbreviation)
Pyruvate and G3P	2.2.1.7	1-deoxy-D-xylulose-5-phosphate synthase (DXS)	1-deoxy-D-xylulose-5-phosphate (DXP)
DXP and NADPH	1.1.1.267	DXP reductorisomerase (DXR)	methylerythritol-phosphate (MEP)
MEP	2.7.7.60	MEP cytidylyltransferase (MCT)	4-(cytidine 5′-diphospho)-2-*C*-methyl-D-erythritol (CD-ME)
CD-ME and ATP	2.7.1.148	CD-ME kinase (CMK)	4-difosfocitidil-2-C-metil-Deritritol 2-fosfato (CD-MEP)
CD-MEP	4.6.1.12	2*C*-methyl-D-erythritol-2,4-cyclodiphosphate synthase (MDS)	2*C*-methyl-D-erythritol-2,4-cyclodiphosphate (MEC)
MEC and NADPH	1.17.7.3	1-hydroxy-2-methyl-2-(*E*)-butenyl 4-diphosphate synthase (HDS)	1-hydroxy-2-methyl-2-(*E*)-butenyl 4-diphosphate (HMBPP)
HMBPP and NADPH	1.17.7.4	HMBPP reductase (HDR)	isopentenyl pyrophosphate (IPP)
IPP	5.3.3.2	isopentenyl diphosphate isomerase (IDI)	dimethylallyl pyrophosphate (DMAPP)
IPP and DMAPP	2.5.1.1	GPP synthase (GPPS) **	geranyl diphosphate (GPP)
GPP and IPP	2.5.1.10	FPP synthase (FPPS) **	farnesyl diphosphate (FPP)
FPP and IPP	2.5.1.29	GGPP synthase (GGPPS) **	geranylgeranyl diphosphate (GGPP)

* Kyoto Encyclopedia of Genes and Genomes (https://www.genome.jp/kegg/, accessed on 1 March 2023). ** GPPS, FPPS, and GGPPS are polyprenyl synthase family (PPS) enzymes.

**Table 2 microorganisms-13-02528-t002:** Comparative genomic features of the ten SDF strains analysed.

Strain	Size (bp)	Scaffold	N50 (bp)	GC Content (%)	CDS #	Protein Coding Regions	Pseudo Genes (Total)	rRNA Genes (5S; 16S; 23S)	tRNA Genes	GenBank Accession
*Lysinibacillus fusiformis* SDF0005	4,472,771	24	392,231	37.6	4369	4328	41	13; 7; 7	85	* VKHW00000000.1
*Bacillus pumilus* SDF0011	3,686,817	56	143,274	41.2	3688	3617	71	7; 3; 2	73	* VKHY00000000.1
*Heyndrickxia oleronia* SDF0015	5,267,437	75	151,790	34.7	5127	5018	109	10; 14; 7	129	* VKHZ00000000.1
*Bacillus safensis* SDF0016	3,674,191	25	484,434	41.6	3688	3640	48	4; 1; 1	74	SADW00000000.1
*Peribacillus simplex* SDF0024	5,376,271	45	497,961	40.2	5204	5007	197	14; 7; 6	81	* VKHX00000000.1
*Paenibacillus popilliae* SDF0028	6,580,875	39	611,008	46.5	5684	5519	165	2; 2; 3	62	SADY00000000.1
*Lysinibacillus sphaericus* SDF0037	5,122,785	71	215,682	36.5	4869	4643	226	5; 7; 2	71	SADV00000000.1
*Brevibacillus brevis* SDF0063	6,239,737	31	471,412	47.3	5789	5602	187	1; 16; 9	89	SADX00000000.1
*Bacillus velezensis* SDF0141	3,945,527	15	962,078	46.4	3887	3780	107	8; 3; 2	78	* VKIB00000000.1
*Bacillus velezensis* SDF0150	3,927,067	21	271,062	46.4	3870	3763	107	8; 6; 2	82	* VKIC00000000.1

* This work.

**Table 3 microorganisms-13-02528-t003:** Occurrence of the polyprenyl synthase family across the ten SDF strains analysed.

Strain	Occurrence	Identity (%) *	Reference Species	GeneBank Reference Sequence
*Lysinibacillus fusiformis* SDF0005	+	99.66	*Lysinibacillus fusiformis*	KAB0443654.1
*Bacillus pumilus* SDF0011	−	NA	NA	NA
*Heyndrickxia* oleronia SDF0015	+	67.86	*Bacillus pumilus*	WP_268443628.1
*Bacillus safensis* SDF0016	−	NA	NA	NA
*Peribacillus simplex* SDF0024	+	98.65	*Peribacillus* sp.	WP_241589686.1
*Paenibacillus popilliae* SDF0028	−	NA	NA	NA
*Lysinibacillus sphaericus* SDF0037	−	NA	NA	NA
*Brevibacillus brevis* SDF0063	−	NA	NA	NA
*Bacillus velezensis* SDF0141	+	100	*Bacillus velezensis*	ASK59031.1
*Bacillus velezensis* SDF0150	+	99.65	*Bacillus velezensis*	QWC45887.1

* The highest identity level from the alignments between each SDF amino acid sequence obtained by in silico translation and the NCBI reference sequence obtained by Blast. The symbols indicate the presence (+) or absence (−) of the corresponding coding gene in the genomes. NA: Not applicable.

**Table 4 microorganisms-13-02528-t004:** Gene coding for the TS enzymes identified in the ten SDF strains analysed using antiSMASH.

Strain	Gene/TS Enzyme		
	sqhC/SHC	Phytoene and/or Squalene Synthase Family Gene/PSS	crti/PDS
*Lysinibacillus fusiformis* SDF0005	−	+	−
*Bacillus pumilus* SDF0011	+	+	+
*Heyndrickxia* oleronia SDF0015	+	−	−
*Bacillus safensis* SDF0016	+	+	+
*Peribacillus simplex* SDF0024	+	+	−
*Paenibacillus popilliae* SDF0028	+	−	−
*Lysinibacillus sphaericus* SDF0037	−	+	−
*Brevibacillus brevis* SDF0063	+	−	−
*Bacillus velezensis* SDF0141	+	+	−
*Bacillus velezensis* SDF0150	+	+	−

The symbols indicate the presence (+) or absence (−) of the corresponding gene sequence coding for a TS in the genomes.

## Data Availability

The data presented in this study are openly available in [GeneBank] at [https://www.ncbi.nlm.nih.gov/genbank/about/, accessed on 1 March 2023], reference number [VKHW00000000.1 VKHY00000000.1 VKHZ00000000.1 SADW00000000.1 VKHX00000000.1 SADY00000000.1 SADV00000000.1 SADX00000000.1 VKIB00000000.1 VKIC00000000.1].

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
