# Peer review of "Genome Mining Reveals Pathways for Terpene Production in Aerobic Endospore-Forming Bacteria Isolated from Brazilian Soils"

_microorganisms, 2025, doi:10.3390/microorganisms13112528_

Round 1
Reviewer 1 Report
Comments and Suggestions for Authors
This study investigates terpene biosynthesis potential in 10 AEFB strains using genome mining. The authors identify key genes in the MEP pathway, polyprenyl synthase family, and terpene synthases, revealing unique biosynthetic gene clusters. Phylogenetic analysis suggests terpene production is an ancestral trait in AEFB. The findings highlight genome mining as a tool for discovering bioactive compounds, though experimental validation is needed to confirm terpene production.
- To strengthen the Introduction section, you should expand on previous research regarding terpene biosynthetic pathways in anaerobic fungi and bacteria (AEFB). Consider including: key studies, findings, comparison and research Gaps What are the parameters for gene annotation in the methods section?
- Please review the Methods section and ensure that all necessary parameters are included. Key parameters to check (e.g. E-value threshold for BLASTp)
- Why used Pearson’s correlation for similarity analysis?
- In section 3.5: How do these enzyme distribution patterns correlate with known phylogenetic relationships among the strains?
- Could the heatmap be supplemented with additional analyses (e.g., principal component analysis) to further clarify enzyme distribution patterns?
- In section 3.6. Were any structural predictions or functional domain analyses performed to confirm that these sequences encode active terpene synthases?
- How do these findings compare to well-characterized bacterial terpene biosynthesis pathways, such as those in Streptomyces or Cyanobacteria?
- Does the variability in hopanoid biosynthesis gene arrangements suggest past horizontal gene transfer events?
- The study suggests industrial applications—how do these AEFB-derived terpenes compare in. yield or efficiency with established microbial production systems?
- Since the findings are based on predictions, what would be the next steps in experimentally validating terpene production in these strains? Do you have any plan to use RNA experiment to verify what you found in this study? Please include a discussion of future work.
Need to be improved.
Author Response
Please see a PDF

Reviewer 2 Report
Comments and Suggestions for Authors
Peer Review of the Manuscript: 'Genome mining reveals pathways for terpene production in aerobic endospore-forming bacteria isolated from Brazilian soils'
Introduction: Some sections contain excessive detail on general terpene biosynthesis pathways that could be condensed. Moreover, the transition to the study’s specific focus on Brazilian soils feels abrupt. Please, condense the description of terpene pathways and shift focus earlier to the rationale for selecting Brazilian soil samples and AEFB.
Methods and results: The methodology is well-documented with sufficient detail for reproducibility. The results are well-organized and supported by appropriate figures and tables.
Discussion: The discussion effectively relates the findings to previous studies and emphasizes the potential of AEFB for terpene production. But it lacks discussion on the limitations of the study, particularly regarding bioinformatics predictions and in vitro validation.
Conclusion: The conclusion succinctly summarizes the main findings and their implications for terpene biosynthesis in AEFB. However, specific experimental approaches for validating the biosynthetic pathways identified in this study should be incorporated.
Reviewer 3 Report
Comments and Suggestions for Authors
Title: Genome mining reveals pathways for terpene production in aerobic endospore-forming bacteria isolated from Brazilian soils
This study investigates the biosynthetic potential of aerobic endospore-forming bacteria (AEFB) for terpene production, utilizing genome mining approaches. The research presents valuable insights into the diversity of biosynthetic gene clusters (BGCs) in AEFB strains and their potential ecological and biotechnological applications. However, several key aspects need to be improved to strengthen the study’s scientific rigor and clarity.
The following are my main comments:
The second paragraph should better contextualize the ecological and industrial roles of terpenes. Currently, the text lacks depth and relevant examples. I suggest incorporating additional examples of terpenes’ roles in plant-microbe interactions, quorum sensing, antimicrobial activity, and industrial applications (e.g., pharmaceuticals, biofuels, and flavoring agents).
In the third paragraph, there are errors in citation formatting, specifically after citations [13] and [17]. Please carefully check and correct these issues.
The relationship between AEFB and terpene biosynthesis is not clearly explained. Since this is a fundamental aspect of the study, the authors should explicitly address whether terpenes have a known ecological function in AEFB. If not, a hypothesis or rationale for studying terpene biosynthesis in these bacteria should be provided.
The sampling strategy is unclear. Why were these specific locations chosen? What is the ecological or industrial relevance of studying AEFB from Brazilian soils?
The strain selection process needs further clarification. How were the 10 strains chosen, and what criteria were used? Was there any consideration of phylogenetic diversity or functional potential in the selection?
The authors need to provide species identification data for the isolated strains. Additionally, a phylogenetic analysis comparing these strains to closely related species should be included to contextualize their evolutionary relationships.
The BGC analysis from antiSMASH should be fully presented. Specifically: How do the BGCs of the six selected bacterial strains compare? How different are the two B. velezensis strains in terms of BGC composition and organization? This could provide insights into strain-level diversification.
The analysis of PPS enzymes (involved in terpene biosynthesis) is too limited. Given their biological relevance, it is essential to assess their homology across taxonomic levels (phylum, family, genus). A comparative sequence analysis, possibly including structural modeling or evolutionary conservation, would strengthen this section.
Table 3: There is a typographical error in the footnote ("in silico" is incorrectly formatted). The identity data in Table 3 lacks context. How was the reference species selected, and why is this comparison meaningful? The rationale for including these identity values should be clearly explained.
The evolutionary analysis of BGCs is insufficient. It is not enough to analyze only key enzyme genes. To provide a comprehensive understanding, the following should be considered: Gene synteny should be evaluated to determine conservation and rearrangements across strains.
Accessory genes within the BGCs should be analyzed, as they may contribute to functional diversity and strain adaptation.
A network-based approach (e.g., BiG-SCAPE) is strongly recommended to cluster BGCs and infer evolutionary trajectories.
Strain diversification should be explicitly addressed, particularly regarding BGC variability. Does BGC variation correlate with strain phylogeny or ecological factors?
Finally, ensure consistent citation formatting and correct parentheses placement in the text. Improve figure/table captions to enhance clarity.
Round 2
Reviewer 1 Report
Comments and Suggestions for Authors
No further comments.
Author Response
Thanks for this recommendation. Based on your suggestion, the English language has been revised.
Reviewer 3 Report
Comments and Suggestions for Authors
Thank you for the revised version of the manuscript. I find the current version satisfactory and appreciate the improvements made. My only suggestion is to consider moving the second paragraph of the Conclusion section to the end of the Discussion, or to another suitable place within the Discussion where it fits more naturally. This would help maintain a clearer distinction between the interpretation of results and the final summary of the study.
Author Response
Comment: Thank you for the revised version of the manuscript. I find the current version satisfactory and appreciate the improvements made. My only suggestion is to consider moving the second paragraph of the Conclusion section to the end of the Discussion, or to another suitable place within the Discussion where it fits more naturally. This would help maintain a clearer distinction between the interpretation of results and the final summary of the study.
Response: We appreciate the recognition and kind words about our work. Moving the second paragraph from the Conclusion section (page 19) to the end of the Discussion section (highlighted in red) on the same page has improved our paper, and we are grateful for this additional suggestion. Additionally, we have fused the first and third paragraphs of the Conclusion section into one.